# Comprehensive Analysis of FASN in Tumor Immune Infiltration and Prognostic Value for Immunotherapy and Promoter DNA Methylation

**DOI:** 10.3390/ijms232415603

**Published:** 2022-12-09

**Authors:** Mingyang Zhang, Lei Yu, Yannan Sun, Li Hao, Jing Bai, Xinyu Yuan, Rihan Wu, Mei Hong, Pengxia Liu, Xiaojun Duan, Changshan Wang

**Affiliations:** School of Life Science, Inner Mongolia University, Hohhot 010020, China

**Keywords:** FASN, immune infiltration, immunotherapy, DNA methylation, miRNA, pan-cancer

## Abstract

Fatty acid synthase (FASN) promotes tumor progression in multiple cancers. In this study, we comprehensively examined the expression, prognostic significance, and promoter methylation of FASN, and its correlation with immune cell infiltration in pan-cancer. Our results demonstrated that elevated FASN expression was significantly associated with an unfavorable prognosis in many cancer types. Furthermore, FASN promoter DNA methylation can be used as a tumor prognosis marker. Importantly, high levels of FASN were significantly negatively correlated with tumor immune infiltration in 35 different cancers. Additionally, FASN was significantly associated with tumor mutational burden (TMB) and microsatellite instability (MSI) in multiple malignancies, suggesting that it may be essential for tumor immunity. We also investigated the effects of FASN expression on immunotherapy efficacy and prognosis. In up to 15 tumors, it was significantly negatively correlated with immunotherapy-related genes, such as PD-1, PD-L1, and CTLA-4. Moreover, we found that tumors with high FASN expression may be more sensitive to immunotherapy and have a good prognosis with PD-L1 treatment. Finally, we confirmed the tumor-suppressive effect of mir-195-5p through FASN. Altogether, our results suggested that FASN may serve as a novel prognostic indicator and immunotherapeutic target in various malignancies.

## 1. Introduction

The immunotherapy development for malignancies has brought about a paradigm shift over the past decade. Although immunotherapy has brought benefits for many patients, it continues to fail because of the intricate tumor microenvironment and the metabolic reprogramming in tumor cells. The aberrant accumulation of fatty acids and cholesterol in tumor-infiltrating myeloid cells, such as myeloid-derived suppressor cells (MDSCs), dendritic cells (DCs), and tumor-associated macrophages cells (TAMs), leads to these immune cells exhibiting immunosuppressive and anti-inflammatory phenotypes through metabolic reprogramming; hence, metabolic interventions are expected to improve the efficacy of immunotherapy [1].

Currently, a number of studies have demonstrated that FASN is highly expressed in a wide range of tumor cells, through the de novo synthesis of fatty acids to satisfy the high-energy requirements of tumor cells and promote malignant tumor progression, including tumor proliferation, invasion, angiogenesis, and metastasis to other lesions [2,3,4,5]. Additionally, FASN connects the metabolism of immune cell control and the tumor microenvironment. For instance, FASN absence in T reg cells can prevent tumor cell proliferation. Researchers have noted that reprogramming lipid metabolism improves T reg cells’ function in tumors, suggesting a potential novel cancer treatment strategy [6]. Furthermore, clinical data from patients with ovarian cancer showed that immunosuppression is frequently associated with increased FASN expression. Li et al. demonstrated that decreasing FASN activity could improve T cell anti-tumor immunity while partially restoring the antigen-presenting ability of tumor-infiltrating dendritic cells, thereby contributing to ovarian cancer therapy [7]. Furthermore, Shen et al. demonstrated that FASN inhibition can reduce the expression of PD-1 receptor ligand in non-small-cell lung cancer cells via the “FASN-TGFβ1-PD-L1” axis, and lead to the cytotoxicity of drug-resistant cells to natural killer (NK) cells [8]. These data implied that targeting FASN may improve immunotherapy by altering the tumor immunological microenvironment.

Currently, several specific FASN inhibitors have been developed as prospective therapeutic targets for a variety of malignancies. FASN inhibitors, including cerulenin and C75, have been extensively investigated in preclinical oncology [9,10,11]. C75 could suppress tumor growth to regulate the function of CD8+ T lymphocytes in the tumor microenvironment by serving as the specific inhibitor of PI3Ks [12]. Another inhibitor of FASN, orlistat, can suppress tumors by increasing CD8+ T cells and reducing T reg cells in a mouse melanoma model, demonstrating that FASN may be involved in tumor immunity regulation [13]. However, the drug toxicity and side effects limit its clinical application [14]. Although the new FASN inhibitors TVB-2640, TVB-3664, TVB-3166, TVB-3664, and TVB-3166 showed high specificity and superior anti-tumor activity as specific FASN inhibitors [15,16,17], their clinical research is still in the early stages, and have not been broadly applied in tumor therapy. Therefore, the use of FASN-targeting miRNAs to construct pharmaceuticals is a plausible strategy to develop new small-molecule medications, given the beneficial features of miRNAs, namely low immunogenicity and harmless or adverse effects of degradation.

However, the function of FASN in various human malignancies and its correlation with tumor immunity have not been well characterized. Here, we examined the expression profile and prognostic value of FASN in various malignancies, as well as the relationship between its expression and immune infiltration, immunological checkpoints, TMB, MSI, prognosis, and responsiveness for immunotherapy. Finally, mir-195-5p capable of disrupting FASN function in hepatocellular carcinoma has been identified. Collectively, these findings suggest that FASN is likely a reliable tumor prognostic and immunotherapy marker.

## 2. Results

### 2.1. Differential Expression of FASN between Tumor and Normal Tissue Samples

It has been reported that FASN plays a substantial role in normal cells and is involved in tumor progression by regulating energy metabolism, which is a possible target for cancer therapy [2,18,19]. First, FASN expression was examined in several tissues under normal physiological settings using the GTEx (genotype-tissue expression) dataset. We found that FASN is more abundantly expressed in adipose tissue, breast tissue, and skin than that in kidney, muscle, heart, and blood tissue (Figure 1A).

We then analyzed the levels of FASN in distinct malignancies. Prostate cancer, liver cancer, and breast cancer had the highest levels of FASN expression, whereas kidney renal clear cell carcinoma (KIRC) and kidney chromophobe (KICH) exhibited the lowest levels of FASN expression, which is similar to the expression profile of FASN in normal tissues (Figure 1B). In tumor tissues, FASN was expressed at a higher level than that in the corresponding control tissues, including glioma (GBMLGG), brain lower grade glioma (LGG), uterine corpus endometrial carcinoma (UCEC), cervical squamous cell carcinoma and endocervical adenocarcinoma (CESC), esophageal carcinoma (ESCA), stomach and esophageal carcinoma (STES), kidney renal papillary cell carcinoma (KIRP), pan-kidney cohort (KIPAN), colon adenocarcinoma (COAD), colon adenocarcinoma/rectum adenocarcinoma esophageal carcinoma (COADREAD), prostate adenocarcinoma (PRAD), liver hepatocellular carcinoma (LIHC), high-risk Wilms tumor (WT), bladder urothelial carcinoma (BLCA), rectum adenocarcinoma (READ), ovarian serous cystadenocarcinoma (OV), pancreatic adenocarcinoma (PAAD), testicular germ cell tumors (TGCT), uterine carcinosarcoma (UCS), acute lymphoblastic leukemia (ALL), acute myeloid leukemia (LAML) (*p* < 0.001), stomach adenocarcinoma (STAD), and head and neck squamous cell carcinoma (HNSC) (*p* < 0.01), but expression was significantly decreased in lung adenocarcinoma (LUAD), lung squamous cell carcinoma (LUSC), thyroid carcinoma (THCA) (*p* < 0.001), glioblastoma multiforme (GBM), and breast invasive carcinoma (BRCA) (*p* < 0.01) in contrast to normal tissues (Figure 1C). Using the CPTAC proteome database, we also examined the differences in FASN protein expression between various tumor tissues and comparable normal control tissues. Consistent with the TCGA database expression trend, FASN expression was significantly higher in COAD, UCEC, and PAAD, and demonstrated a decreasing trend in comparison with normal samples in BRCA, ovarian cancer, GBM, liver cancer, and lung cancer (Figure 1D). Overall, FASN was significantly increased in twenty-three (total thirty-four) cancer types and was markedly decreased in five cancer types based on the TCGA and GTEx data. Moreover, the twenty-three types of cancer, include almost all the high-incidence and high-mortality cancer types, except breast cancer [20], which further illustrated the carcinogenesis of FASN in the tumor progression.

### 2.2. The Correlation of FASN Expression and Tumor Grade or Stage

We examined the relationships between FASN expression and the pathological grades and stages of many tumor types, given FASN expression is altered following the initiation of numerous highly lethal cancer types. The results demonstrated that the overall trend of FASN expression was increased in CESC (*p* < 0.05), KIRC (*p* < 0.01), and UCEC (*p* < 0.001) with increasing tumor malignancy (i.e., lower tumor differentiation), but decreased in HNSC and LGG (*p* < 0.001), decreasing with tumor progression (Figure 2A). Furthermore, the tumor stage results further demonstrated that FASN is involved in the regulation of tumor progression. In KIRP (*p* < 0.001), KIRC (*p* < 0.001), TGCT (*p* < 0.01), adrenocortical carcinoma (ACC, *p* < 0.05), BRCA (*p* < 0.05), and CESC (*p* < 0.05), the expression of FASN was increased with the stage of tumor progression and was negatively correlated only in OV (*p* < 0.01) (Figure 2B). Its expression was not correlated with tumor grade and stage in BRCA, LUAD, LUSC, and THCA with low FASN expression, while in CESC, KIRP, TGCT, and UCEC with high FASN expression, FASN expression tended to increase significantly in tumor grade or stage with tumor progression when we took into account the changes of FASN expression in the tumor versus normal tissue (Appendix A). In particular, FASN expression in CESC showed a significant increase in both stage and grade, and Shang et al. demonstrated that targeting FASN by the combination of miR-532-5p and the FASN inhibitor orlistat significantly inhibited lymph node metastasis in cervical cancer mouse xenografts [21]. Moreover, Du et al. also showed that FASN can promote lymph node metastasis in cervical cancer through cholesterol re-programming and lymphangiogenesis [22], further demonstrating that FASN was associated with the malignant progression of CESC. Overall, we speculated that up-regulation of FASN in tumors maybe promote tumor progression.

### 2.3. Prognostic Analysis of FASN in Patients with Different Cancers

To further investigate the relationship between FASN expression and tumor prognosis, we employed Sangerbox3.0 to examine FASN expression for overall survival in 44 cancer types and disease-free interval in 32 cancer types. The observations indicated that high FASN expression was not only associated with poor overall survival in CESC (*p* = 8.5 × 10^−3^, HR = 1.39), SARC (*p* = 4.1 × 10^−4^, HR = 1.58), KIRP (*p* = 1.8 × 10^−5^, HR = 1.83), KIPAN (*p* = 1.7 × 10^−7^, HR = 1.41), KIRC (*p* = 3.7 × 10^−7^, HR = 1.48), BLCA (*p* = 5.4 × 10^−3^, HR = 1.24), mesothelioma (MESO, *p* = 1.5 × 10^−3^, HR = 1.55), ACC (*p* = 4.8 × 10^−3^, HR = 1.36), LAML (*p* = 0.02, HR = 1.20), and KICH (*p* = 0.02, HR = 3.73), but also with poor disease-free interval in CESC (*p* = 0.02, HR = 1.70), KIRP (*p* = 4.5 × 10^−4^, HR = 1.90), KIPAN (*p* = 0.01, HR = 1.38), and ACC (*p* = 8.5 × 10^−4^, HR = 1.77) (Figure 3A,B), ultimately indicating that high FASN expression in CESC, KIRP, KIPAN, and ACC will predict poor prognosis. The disease-free interval results for FASN indicated that high expression was a protective factor only in OV (*p* = 0.02, HR = 0.80) (Figure 3B). Consequently, high FASN expression may be a prognostic risk factor for CESC, KIRP, KIPAN, and ACC patients, whereas high FASN expression may be a protective factor for OV patients.

### 2.4. FASN Expression Regulates Tumor Infiltration of Immune Cells in Multiple Cancers

As a comprehensive understanding of the functions of FASN, it has been discovered that FASN participates in the regulation of the tumor microenvironment as a metabolic regulator, which in turn affects immune cell infiltration or treatment sensitivity. Targeting FASN enhances the body’s anti-tumor function through immune-regulation-dependent pathways [6,7,12,23]. Therefore, we evaluated the correlation between FASN expression and immunity in pan-cancer. The relationships between FASN expression and tumor stromal, immune, and ESTIMATE scores were examined using ESTIMATE. There was a strong negative correlation between the expression of FASN and immune infiltration (Appendix A) (Stromalscore: 33/44 cancer types were significantly negatively correlated, and one was positively correlated; Immunescore: 34/44 were significantly negatively correlated; ESTIMATEscore: 35/44 were significantly negatively correlated). The top four significant correlation cancers were GBMLGG, LGG, STES, and STAD (Stromalscore) (Figure 4A); GBMLGG, LGG, ESCA, and STES (Immunescore) (Figure 4B); and GBMLGG, LGG, ESCA, and STES (ESTIMATE scores) (Figure 4C). Additionally, the ImmuCellAI results revealed a strong negative correlation between FASN expression and the infiltration scores in 18 cancer types across 33 cancer types. Moreover, immune infiltration of CD4+ T cells, CD8+ T cells, cytotoxic T cells, NK cells, Tfh T cells, and Th2 T cells was significantly negatively correlated with FASN expression in the majority of malignancies (Figure 4D, the makers of immune cell population were shown in Appendix A). Moreover, we found that FASN expression was significantly positively correlated with monocytes, M0 macrophages in a variety of cancers (STAD, SKCM, SARC, OV, LUSC, LUAD, KIRC, HNSC, CHOL), and that M2 macrophages, which normally exert anti-inflammatory effects in tumors, were also significantly positively correlated with THYM, THCA, TGCT, KIPAN, BRCA, (Figure 4D,E), suggesting that FASN may be involved in the process of monocyte-to-macrophage differentiation and M2-macrophage-mediated anti-inflammatory effects in tumors. FASN expression was significantly negatively correlated with CD8+ T cells (BLCA, BRCA, CESC, HNSC, LGG, LUAD, LUSC, OV, SKCM, STAD), CD4+ T cells (BLCA, BRCA, ESCA, HNSC, PRAD, SARC, SKCM, STAD, TGCT, THCA), and DC cells (BLCA, HNSC, LGG, LIHC, LUAD, LUSC) infiltration in multiple cancer cell types using TIMER analysis, which is consistent with the ImmuCellAI results (Figure 4F).

Additionally, we discovered that in up to fourteen malignancies, FASN expression was significantly associated with tumor immune subtypes (Appendix A). To further investigate the regulatory function and molecular mechanism of FASN in the tumor microenvironment, we analyzed the MSI and TMB of FASN. The results showed a strong correlation between FASN expression and MSI in twelve cancers. In eleven tumors (GBM, GBMLGG, CESC, LUAD, STES, KIPAN, STAD, TGCT, LUSC, UVM, and BLCA), significant positive relationships were discovered (Figure 5A).

For TMB, FASN expression was positively correlated with CESC, LUAD, STES, SARC, KIPAN, STAD, KIRC, READ, and CHOL (Figure 5B). The main cause of MSI was the genetic variation or epigenetic alteration of DNA mismatch repair (MMR); therefore, we analyzed the main members of the MMR system, MutL homolog 1 (MLH1), MutS protein homolog 2 (MSH2), MutS Homolog 6 (MSH6), and PMS1 homolog 2 (PMS2) and found that they were associated with FASN expression, and the results revealed that the FASN and MMR genes were significantly positively correlated with 29 cancers (Figure 5C).

Surprisingly, almost all major histocompatibility complex molecules (MHCs), except TAP2, were negatively correlated with FASN expression in a variety of malignancies, especially in LGG (Figure 5D), when we examined the correlation between FASN expression and MHCs in pan-cancer. We speculated whether FASN exhibited negative correlations in multiple cancer types directly by affecting the transcription factors shared by these MHC molecules. We then used TRRUST to predict the potential upstream transcription factors of all MHCs and found that CIITA, RFX5, RFXANK, and RFXAP were the most likely transcription factors (Appendix A), and the expression of FASN was significantly negatively correlated with MHC class II transcriptional activators CIITA and RFXANK in a variety of cancers (Figure 5E), which is consistent with the findings in MHCs. Additionally, the CIITA prognostic results showed that its high expression in four tumor types was associated with a poor prognosis, and its low expression in eight tumor types was associated with a poor prognosis (Appendix A). Similarly, NLRC5, a key transcription factor of MHC class I, was significantly correlated with FASN in a variety of tumors, for example, FASN was significantly negatively correlated with HLA-A, HLA-B, HLA-C, HLA-E, HLA-F, HLA-G, and NLRC5 in SARC, while it was significantly positively correlated in COAD (Figure 5E). Therefore, we hypothesized that FASN might regulate MHC class I and II molecules through NLRC5 and CIITA, respectively. In addition, we investigated the relationship between FASN and immune checkpoint gene expression and found that FASN was significantly positively correlated with the immune checkpoint genes CD276 and VEGFA in as many as 27 cancer species (Appendix A). Simultaneously, we found that the expression of FASN was correlated with various chemokines (e.g., XCL2 and CCL14) and chemokine receptors (e.g., XCR1 and CCR8) in different cancers (Appendix A). These findings further suggested that FASN is essential for controlling pan-cancer immune cell infiltration.

### 2.5. FASN Offers Significant Prognostic Value for Immunotherapy

The aforementioned findings motivated us to investigate the impact of FASN on immunotherapy, and we found that FASN expression was significantly negatively correlated with key targets of immune checkpoint blockade therapy, including programmed death-1 (PD-1), programmed death ligand-1 (PD-L1), and cytotoxic T lymphocyte antigen-4 (CTLA-4), in up to 15 cancer types [24], especially in LGG (Figure 6A). Additionally, high levels of FASN predicted a poor prognosis following anti-PD-1 therapy according to Kaplan–Meier plotter analysis (Figure 6B), which was consistent with the prognosis of GBM patients receiving anti-PD-1 (Figure 6C). On the contrary, the high expression of FASN had a positive effect on the prognosis of anti-PD-L1 patients (Figure 6D), which may be due to the specific effect of FASN on tumor cells and tumor-infiltrating immune T cells, which requires further study. In both anti-PD-L1 immunotherapy of BLCA and anti-CTLA-4 immunotherapy of melanoma, high expression of FASN predicted a poor prognosis (Figure 6E,F). High FASN expression was significantly associated with poor prognosis with anti-PD-1 combined with anti-CTLA-4 treatment. Additionally, we discovered that there was a statistically significant difference in the expression of FASN between the response and non-response groups treated with anti-PD-1 and anti-CTLA-4 (Figure 6G,H), indicating that FASN may be directly related to the outcome of immunotherapy. Ultimately, we identified a pattern in which patients with high FASN expression tended to be more responsive to immunotherapy, which was observed in the melanoma immunotherapeutic cohort among the four cohorts evaluated (Figure 6H–K). The analysis results demonstrated that the effectiveness of immunotherapy is directly correlated with the FASN expression level.

In addition, we examined the FASN mutation status in all types of cancer and found that FASN showed a higher gene mutation frequency in SKCM (13.52%), UCEC (13.24%), and LIHC (10.49%) (Appendix A). Immune infiltration in LIHC, SKCM, and UCEC was significantly influenced by somatic copy number variation (Appendix A). Additionally, CAMOIP data analysis revealed that FASN mutations had a better prognosis than wild-type immunotherapy in patients with melanoma (Appendix A). These findings also suggested that FASN is significantly associated with immune infiltration of various malignancies and that immunotherapy outcomes may be influenced by FASN expression levels and mutation status.

### 2.6. Analysis of DNA Methylation at the FASN Promoter

By modifying the accessibility of chromatin, DNA methylation not only controls gene expression but also participates in a variety of biological processes in normal cells [25,26]; however, its abnormal changes in cancer are also thought to be promising candidates for the development of prognostic, diagnostic, and predictive biomarkers [27]. We used the MethSurv database to analyze the distribution of DNA methylation at the FASN loci in the high-expressing tumors in the GTEx and TCGA databases to ascertain whether the high expression of FASN in numerous cancers (BLCA, CESC, KIRP, STAD, KIRC, ESCA, HNSC, UCS, READ, COAD, UCEC, and LIHC) was regulated by promoter DNA methylation. Seven probes close to the promoter were DNA hypomethylated in all tumors with increased FASN expression (Figure 7A, S4A–F, and S5A–E). We speculated that the high expression of FASN was caused by hypomethylation of the FASN promoter. We further examined eleven cancer-type DNase-seq data from ENCODE and discovered that they were significantly enriched in FASN promoters (Figure 7B). Therefore, we preliminarily demonstrated that FASN promoter hypomethylation leads to chromatin opening, possibly explaining why FASN is highly expressed in various tumors. Further analysis revealed that the seven probes were significantly different in ten malignancies, including BLCA, BRCA, COAD, HNSC, KIRC, LIHC, LUSC, PAAD, PRAD, and THCA (Figure 7C).

Not all CpG islands function equally in controlling gene expression, even within a single promoter region, and that specific region rather than the entire promoter region generally influences transcriptional silencing [27]. When we examined the CpG island function of probes closer to the TSS (TSS200: cg24715260), we discovered that in BRCA, FASN expression and methylation levels were strongly inversely correlated, but only considerably favorably correlated in LUSC (Figure 8A). This is consistent with the universally recognized effect of promoter hypermethylation on transcriptional suppression. In addition, we examined the survival association of these seven probes in the ten cancer types and discovered that, apart from PRAD and THCA, methylation of the probe site was significantly correlated with patient survival in the other eight malignancies (Figure 8B). These results suggested that promoter methylation of FASN has important prognostic value for tumors.

### 2.7. MiRNA-Targeting FASN Prediction and Tumor Suppressor Function Verification

Scientists have recently embraced RNA-targeted molecular therapeutics owing to their precise targeting, and since miRNAs are naturally occurring molecules in the human body, there is potential for the development of therapeutics [28]. Therefore, we identified and analyzed miRNAs that might target FASN. First, we obtained miRNAs from miRDB, TargetScan, ENCORI, and mirDIP that may target FASN. For prediction accuracy, we took the intersection of four database miRNAs and obtained seven miRNAs that may target FASN, as shown in Figure 9A.

FASN is highly expressed in most cancers, and its high expression indicated poor prognosis. In contrast, given the negative regulatory mechanism of “miRNA-Target” [29], we chose mir-195-5p and mir-497-5p, which have a negative correlation with FASN and are typically decreased in cancer for further investigation (Appendix A). Mir-195-5p significantly decreased in BRCA, UCEC, LUAD, STAD, THCA, LUSC, LIHC, ESCA, HNSC, BLCA, KICH, KIRP (*p* < 0.001), and CESC, PCPG (*p* < 0.01) compared to that in healthy tissues, as shown in Figure 9B. Additionally, mir-195-5p was significantly negatively correlated with FASN in TGCT, STAD, UCEC, SARC, CESC, ESCA, LIHC, BLCA, LUAD, PRAD (*p* < 0.001), KIRC, MESO (*p* < 0.01), KICH, READ and PCPG (*p* < 0.05) according to pearson correlation analysis, with only LGG (*p* < 0.01) and OV (*p* < 0.05) showing positive correlations (Figure 9C). Therefore, we preliminarily determined that mir-195-5p might function by targeting FASN in different cancers. Additionally, mir-497-5p was significantly downregulated in multiple cancers and was significantly negatively correlated with FASN in twelve cancer types (Appendix A). Surprisingly, low expression of mir-195-5p in BRCA (HR = 0.61, *p* = 0.0083), LIHC (HR = 0.63, *p* = 0.0097), LUAD (HR = 0.69, *p* = 0.014), LUSC (HR = 0.71, *p* = 0.042), UCEC (HR = 0.49, *p* = 0.00075), and SARC (HR = 0.6, *p* = 0.011) was significantly associated with poor overall survival, further demonstrating that low expression of mir-195-5p and high expression of FASN are at risk for tumor prognosis (Figure 9D). In addition, the prognostic results of mir-497-5p showed a protective effect in LIHC, CESC, LUAD, and UCEC, and a risk factor in BLCA, KIRC, KIRP, and THCA (Appendix A). Moreover, we found that mir-195-5p and mir-497-5p target genes were enriched in various tumor pathways and critical signaling pathways regulating cell life activities, such as the p53 and PI3K–Akt signaling pathways. Mir-497-5p prognostic analysis showed a protective factor in LIHC, CESC, LUAD, and UCEC, and a risk factor in BLCA, KIRC, KIRP, and THCA (Appendix A). Additionally, we found that the target genes of mir-195-5p and mir-497-5p were abundant in a number of tumor pathways and critical signaling pathways controlling cell life activities, including the PI3K–Akt and p53 signaling pathways (Appendix A). Therefore, the administration of exogenous mir-195-5p and mir-497-5p may be a possible treatment for tumors with high FASN expression, based on the results of the above research.

MiRNA targeting may have a tumor suppressor effect in more malignancies, since prognostic data indicated that low expression of mir-195-5p has a poor prognosis effect in more tumors than mir-497-5p (Figure 9D and Appendix A). Therefore, we selected hepatocellular carcinoma (with relatively high FASN expression) to investigate the inhibitory effect of mir-195-5p in tumors for the next study. First, the expression of mir-195-5p was detected in six pairs of collected HCC samples, and the results indicated that mir-195-5p was significantly lower in five pairs of samples (Appendix A). We also found that mir-195-5p overexpression significantly inhibited the proliferation (Appendix A), clonogenic ability (Appendix A), cell cycle (Appendix A), and invasiveness of hepatoma tumor cell lines (Appendix A). This finding supported the notion that mir-195-5p has a tumor suppressor function.

### 2.8. FASN Functional Enrichment

To gain insight into the regulatory network of FASN in tumor progression, we analyzed the protein interaction of FASN using the GeneMANIA database. The results showed that FASN significantly interacts with the AASDHPPT protein (Figure 10A), and the enrichment results showed that FASN was mainly enriched in pathways such as cellular lipid metabolism (Figure 10B), which was consistent with its role as a fatty acid synthase. To further determine the function of high FASN expression in tumors, we selected three tumors with high FASN expression (BLCA, LIHC, and PRAD) for KEGG enrichment analysis. The findings demonstrated that genes associated with FASN were predominantly enriched in pathways regulating essential functions of tumor cells, such as fatty acid metabolism, non-homologous recombination repair, and other processes. More importantly, in BLCA, LIHC, PRAD, and FASN, negatively related genes were significantly enriched in Th1 and Th2 cell differentiation pathways, and pathways related to tumor immune regulation, such as cytokine–cytokine-receptor interaction, antigen processing and presentation, and chemokine signaling pathways (Figure 10C–E). It is consistent with the above analysis that FASN was significantly negatively associated with immune infiltration in most tumors.

## 3. Discussion

In mammalian tissues, FASN-mediated de novo adipogenesis was limited to adipocytes and hepatocytes, while other tissues were more commonly involved in the direct intake of lipoproteins and free fatty acids from the blood to feed lipid requirements via fatty acid translocase CD36, or fatty acid transport protein family [30]. In contrast, the lipid de novo synthesis pathway is abnormally activated when cancer occurs, and researchers have demonstrated that this mode provides most of the lipids for cancer cell proliferation and promotes malignant tumor progression [31]. Inhibition of FASN-mediated lipogenesis has been shown to exert powerful anti-tumor effects in a variety of cancers, including breast and ovarian cancers [2,32,33].

FASN is crucial for governing the activity of tumor immune cells [6]; however, its role in pan-cancer and its correlation with tumor immunity have rarely been reported. We firstly assessed FASN expression and its clinical value in the context of cancer and showed that FASN was at a higher level in tumor tissues than that in the control tissues. In GBMLGG, LGG, UCEC, CESC, ESCA, STES, KIRP, KIPAN, COAD, READ, PRAD, LIHC, WT, BLCA, READ, OV, PAAD, TGCT, UCS, ALL, LAML, STAD, and HNSC, it was significantly overexpressed in twenty-three cancer types, whereas significantly low expressed in LUAD, LUSC, THCA, GBM, and BRCA cancer types. We speculated that the differential expression may be involved in the respective tumor microenvironment and major fatty acid uptake patterns of the tumors, for example, leukemic cells can uptake fatty acids from the microenvironment directly using CD36 or the fatty acid binding protein FABP4 in LAML [34,35], and CD36+ LSC can obtain energy directly from the fatty acid oxidation process of adipocytes in the bone marrow microenvironment [36]; these may ultimately lead to a less dependence of LAML on FASN-mediated de novo fatty acid synthesis. However, high expression of FASN is still a risk factor for LAML (Figure 3A). Fhu et al. concluded that FASN promotes progression in a variety of tumors by promoting tumor fat synthesis, signal transduction, and cell migration [33]. Even in the low-expressed LUAD in the TCGA database, the illustrated data show that inhibiting FASN enhances the sensitivity of tumor cells to irradiation [37], further confirming the oncogenic role of FASN in various tumors. In addition, the FASN expression levels were significantly correlated with the clinical grade and stage of various tumors, especially in KIRP, KIRC, TGCT, and CESC, and the expression of FASN was increased with tumor progression. Furthermore, the clinically significant role of FASN in the early diagnosis of these malignancies was indicated by the considerable difference between stage I and stage II tumors.

According to the survival analysis, high FASN expression was related to positive prognosis in GBMLGG, LGG, and PAAD, and poor prognosis in CESC, SARC, KIRP, KIPAN, KIRC, BLCA, MESO, LAML, ACC, and KICH. We also examined the disease-free interval of FASN because non-cancer causes of death impact the overall survival analysis. The data analysis supported the hypothesis that high FASN expression is a risk factor for CESC, KIRP, KIPAN, and ACC but a protective factor for OV. Additionally, high FASN expression has also been related to poor recurrence-free survival in prostate cancer and a poor prognosis in gastric cancer [38,39]. These findings imply that increased FASN expression is a risk factor for carcinogenesis in most tumor types.

The use of immune checkpoint blockade therapy in cancer immunotherapy has recently undergone a paradigm shift. Due to insufficient immune cell activation to recognize tumor-specific antigens in some cancers, immunotherapy is still not frequently employed clinically, even though it restores the autoimmune system’s capacity to recognize and eliminate tumor cells [40,41]. Therefore, it is crucial to identify novel therapeutic targets. Previous studies have demonstrated that CD8 cytotoxic T lymphocytes are the immune cells of preference for battling cancer and are essential for cellular immunity by eliminating tumor cells [42]. By stimulating NK cells, CD8+ T cells, and other innate immune cells, CD4+ T cells can inhibit tumor progression [43], and cytokines produced by CD4+ T cells upon contact with tumor cell surface antigens can stimulate the production and activation of CD8+ T cells [44]. According to our findings, immune infiltration was significantly negatively correlated with FASN expression in up to 35 (totaling 44) cancers. ImmunCellAI research results further demonstrated that in the majority of tumors, the expression of FASN was significantly negatively correlated with CD8+ T cells, CD4+ T cells, cytotoxic T cells, NK cells, Tfh cells, and Th2 cells. Notably, FASN was significantly positively associated with neutrophils in all tumors, except in KICH. In a murine lung cancer model, neutrophils enhanced tumor cell proliferation [45], suppressed NK cell activity, and stimulated tumor metastasis [46]. Therefore, we speculated that high FASN expression in the majority of cancers may enable tumor cells to evade the immune response by preventing the invasion of anti-tumor immune cells. M1 macrophages have been reported to exert pro-inflammatory effects and activate T cell responses, which are associated with a positive clinical prognosis in several tumors, such as non-small-cell lung cancer, and colorectal cancer [47,48,49]. In contrast, M2 macrophages in most tumors are anti-inflammatory and exert immunosuppressive effects in tumors, and typically predict a poor prognosis [50,51,52]. Our results showed a significant positive correlation between FASN and M0 macrophages in a variety of tumors. In THCA, KIPAN, and BRCA, FASN expression was significantly negatively correlated with M1 macrophages and positively correlated with M2 macrophages, suggesting that FASN in these three tumors may promote tumor progression by inhibiting M1 macrophage polarization and promoting M2 macrophage polarization. Our results also showed that FASN was significantly positively correlated with HMGB1 and VEGFA in 31 and 27 cancer species, respectively, and Huber et al. showed that damage-associated molecular pattern high-mobility group box 1 protein (HMGB1) release in the tumor microenvironment would promote the accumulation of M2-polarized macrophages and secretion of IL10 [53], and conversely, malignant tumor cells could secrete M2-associated cytokines, such as IL10 and VEGF, to recruit more monocytes and M0 macrophages to the tumor region and differentiate them into the M2 phenotype [54]; thus, inhibiting FASN may contribute to the formation of a benign tumor microenvironment.

MSI has been reported to act as an independent clinical prognostic marker for colon cancer [55], and TMB is a promising biomarker for immunotherapy [56]. According to our observations, there was a strong positive association between the expression of FASN and MSI in eleven malignancies and TMB in nine tumors. The strong positive association between the expression of FASN and the key MMR system elements in 29 malignancies was also observed, which encouraged us to hypothesize that FASN may be related to genomic instability caused by DNA mismatch repair during carcinogenesis. Our findings further demonstrated a strong correlation between FASN expression and MHCs, 60 immune checkpoint genes, chemokines, and chemokine receptors, suggesting a strong connection between FASN levels and tumor immune regulation. In particular, FASN is significantly negatively correlated with MHCs in most tumors; the primary activity of MHCs participates in antigen presentation by CD4+ T cells and the differentiation or activation of CD8+ T cells [57,58]. This further indicated that FASN is involved in the interaction between tumor cells and immune cells, and the high expression of FASN may play a biological role in anti-tumor immunity in various tumors. Meanwhile, the analysis showed that FASN was significantly positively correlated with CD276 and VEGFA in up to 27 tumors. Yang et al. summarized that antibody targeting the immune checkpoint gene CD276 significantly inhibited CD276-positive tumor cells through antibody-dependent cellular cytotoxicity [59], as enoblituzumab can play a therapeutic role in melanoma. Furthermore, our results showed that high FASN expression tended to be more responsive to immunotherapy in melanoma against different targets (Figure 6H–K). The anti-tumor function of CAR-T cells genetically engineered to target CD276 in glioblastoma was demonstrated in vivo, and in vitro [60]. Additionally, our results indicated that high expression of FASN has a poor prognosis for anti-PD-1 treatment in GBM (Figure 6C), so the selection of immunotherapeutic approaches targeting CD276 in GBM may offer new prospects. Moreover, current anti-angiogenic monoclonal antibodies targeting VEGFA, such as bevacizumab in combination with cancer immunotherapy have clinically benefited patients with non-small-cell lung cancer and hepatocellular carcinoma [61,62], suggesting that tumors with high FASN expression may be more suitable for targeting VEGFA in combination with immunotherapy.

More importantly, FASN expression was significantly negatively correlated with PD-1 in multiple cancers, and its expression has significant prognostic significance for anti-PD-1 and anti-PD-L1 immunotherapy, which has also been validated in immunotherapy prognostic outcomes for GBM. These demonstrated the potential of FASN as a GBM immunotherapy target from a different angle. However, the prognosis of FASN expression in anti-PD-1 immunotherapy and anti-PD-L1 treatment methods is diametrically opposite, which may be due to the different functions of FASN in tumor cells and tumor-associated T cells. However, further experimental studies are required. Additionally, GSEA results demonstrated that in the BLCA, LIHC, and PRAD with high FASN expression, FASN negatively related genes were significantly enriched in Th1 and Th2 cell differentiation, antigen processing and presentation, cytokine–cytokine receptor interaction, and other crucial tumor immune-related pathways. It was consistent with the results of the immune infiltration analysis described above. Collectively, these results suggested that FASN may have a significant immunoregulatory function in malignancies and is an essential factor that influences the effectiveness of immunotherapy.

Here, we investigated the relationship between FASN expression and DNA methylation in human tumors for the first time and found that the promoter of FASN is hypomethylated in almost all cancer types, while the gene body is hypermethylated. The methylation level of the probe (TSS200: cg24715260) was significantly negatively correlated with the expression of FASN in a variety of tumors, and the results suggested that DNA methylation of the probe at the FASN promoter can be used as a prognostic marker for a variety of tumors. Further analytical validation using clinical samples is needed. We searched for FASN-targeting miRNAs as therapeutic candidates because miRNA degradation has no toxic side effects and is relatively simple to modify [63].

We used the mirDB, TargetScan, ENCORI, and mirDIP databases to identify the miRNAs that may target FASN and verified the tumor suppressor effect of mir-195-5p targeting FASN in HCC. The results showed that mir-195-5p had a significant inhibitory effect on the proliferation, cloning, and invasion of tumor cells, consistent with the findings that mir-195-5p inhibited the progression of osteosarcoma and breast cancer by targeting FASN summarized by Yu et al. [64]. In addition, although Duan et al. demonstrated that mir-195-5p is inversely linked with immune cell infiltration in lung adenocarcinoma [65], the precise molecular mechanism has not been investigated. Dobosz et al. showed that miRNAs can also be involved in tumor-cell–immune-cell interactions by regulating post-transcriptional regulation of immune checkpoint genes [66], which made mir-195-5p development as a small molecule drug more challenging. Taken together, these results demonstrated the tumor therapeutic effect of FASN inhibition by mir-195-5p and future studies may take a new direction into the involvement of mir-195-5p and FASN in the tumor immune response.

Although we analyzed the role of FASN in pan-cancer from multiple perspectives, this study had some limitations. There is a systematic bias due to the retrieval of multiple pieces of information from different databases for analysis. Second, the precise mechanism through which FASN is implicated in immune modulation and tumor immunotherapy responsiveness in malignancies has not yet been experimentally validated. It is necessary to explore its function in the immunological control of FASN.

## 4. Materials and Methods

### 4.1. FASN mRNA and Protein Expression Profiles

The gene expression display server (GEDS) database (http://bioinfo.life.hust.edu.cn/web/GEDS/) was used to retrieve mRNA expression data for 30 human normal tissues or organs and 33 FASN cancer types [67]. Sangerbox3.0 tool (http://vip.sangerbox.com) was used to collect the differential expression profile data of FASN in pan-cancers [68]. The Ualcan database (http://ualcan.path.uab.edu/) was used to conduct a pan-cancer investigation of FASN protein expression in ten tumor types.

### 4.2. Survival Analysis and Relationship with Clinical Grade and Stage

The Sangerbox3.0 tool “prognostic analysis” module was used to evaluate the relationship between FASN expression and cancer prognosis. TISIDB (http://cis.hku.hk/TISIDB/index.php) examined the correlation between FASN expression with the grade and stage of several malignancies [69].

### 4.3. Relationship between FASN Expression and Immunity

The ESTIMATE algorithm in the Sangerbox3.0 bioinformatics tool was used to determine the immune and stromal scores for each tumor sample and examine the relationship between scores and FASN expression. The relationship between FASN expression and immune cells in 33 cancer types and immune infiltration scores were examined using the ImmuCellAI algorithm. The CIBERSORT algorithm was used to analyze the correlation of FASN expression and macrophages with different states. The Timer method was used to assess each patient’s B cell, CD4+ T cell, CD8+ T cell, neutrophil, macrophage, and DC infiltration scores in each tumor, and the corr.test function of the R package was used to determine the association between FASN and immune cell infiltration score in each tumor.

The Sangerbox3.0 tool was used to analyze the relationship between FASN expression and MSI and TMB. Timer2.0 was used to evaluate the expression of 60 immune checkpoint genes, MHCs, transcription factors (CIITA, RFX5, RFXANK, RFXAP), chemokines, chemokine receptors, CTLA-4, PD-1, and PD-L1 correlations in pan-cancer. Results were visualized using the SRplot tool (http://www.bioinformatics.com.cn/srplot).

### 4.4. Analysis of Mutations in FASN and Immunotherapeutic Response

The cBioPortal-TCGA pan-cancer panel, which includes 10,953 patients and 10,967 samples from 32 oncology studies, was used to identify genetic FASN aberrations in malignancies. Using the Timer database (https://cistrome.shinyapps.io/timer/), how FASN somatic copy number variants affected immune cell infiltration was established in the top three tumor types with the highest incidence of genetic aberrations. The data (Miao et al., 2018 and Riaz et al., 2018) collected in the CAMOIP database (http://220.189.241.246:13838/) were used to analyze the effect of FASN mutation status on the clinical prognosis of melanoma patients after immunotherapy [70].

The effect FASN expression level on the clinical prognosis of cancer patients following immunotherapy was examined using the “Kaplan-Meier immunotherapy” module of the Kaplan–Meier plotter (http://kmplot.com/analysis/) database. Four sets of immunotherapy patients’ transcriptome data (Gide et al., 2019; Nathanson et al., 2017; Lauss et al., 2017) were gathered from the tumor immune dysfunction and exclusion database (TIDE, http://tide.dfci.harvard.edu). The t-test was then used to identify the differences in FASN expression between the response and non-response groups.

### 4.5. Analysis of miRNAs Targeting FASN

Using four databases (mirDB, http://mirdb.org; TargetScan, http://www.targetscan.org/vert 72/; ENCORI, http://starbase.sysu.edu.cn/; and mirDIP, http://ophid.utoronto.ca/mirDIP/) to predict miRNAs of targeting FASN, seven candidate miRNAs were identified after taking the intersection. The CancerMIRNome (http://bioinfo.jialab-ucr.org/CancerMIRNome/) was used to analyze the expression of miRNA in pan-cancer, the correlation between miRNA and FASN, and the KEGG pathway enrichment of miRNA target genes. The significance of miR-195-5p and mir-497-5p expression on patient prognosis was examined using the Kaplan–Meier plotter.

### 4.6. Correlation between FASN Expression and DNA Methylation

The MethSurv (https://biit.cs.ut.ee/methsurv/) database was used to analyze the distribution of DNA methylation at the FASN locus in pan-cancer and the impact of methylation at individual probe positions on patient prognosis. The GraphPad Prism software (GraphPad, v8.0.2.263, Inc., California, USA) was then used to draw a forest graph. The SMART (http://www.bioinfo-zs.com/smartapp/) database was used to analyze the overall methylation differences of seven probes at the FASN locus and the correlation between DNA methylation levels at the cg24715260 position and FASN expression. Chromatin accessibility near the transcription start site of 11 cancer-type FASNs was analyzed using DNase-seq data collected in the ENCODE (https://www.encodeproject.org/) database and visualized with the integrative genomics viewer (IGV_Win_2.8.10) [71].

### 4.7. Cell Lines and Cell Culture

Huh7 and MHCC97L cells were obtained from Procell Life Science & Technology Co.Ltd. (Wuhan, China) and Zhong Qiao Xin Zhou Biotechnology Co.Ltd. (Shanghai, China), respectively. All cells were cultured in modified Eagle’s medium (DMEM; Biological Industries, Israel) supplemented with 10% fetal bovine serum (FBS; Biological Industries, Israel) in a humidified incubator (5% CO_2_ at 37 °C).

### 4.8. Tissue Sample Sources

Liver tissues were obtained from patients who underwent surgery at the Affiliated Hospital of Inner Mongolia Medical University. Histopathological diagnoses were performed in all patients according to World Health Organization (WHO) criteria. Liquid nitrogen was used to cryopreserve the tissue immediately after surgical excision. The study conformed to the International Ethical Guidelines for Human Biomedical Research, Declaration of Helsinki, and was authorized by the Ethics Committee of the Affiliated Hospital of Inner Mongolia Medical University. Informed consent was obtained from all the participants.

### 4.9. MiRNA Detection and Transfection

RNA was extracted from tissue and cell samples using RNAiso (TAKARA, Beijing, China) according to the manufacturer’s instructions and reverse-transcribed using the Mir-X miRNA first-strand strand synthesis system (TAKARA, Beijing, China). Finally, quantitative PCR was used to detect the relative expression of mir-195-5p in multiple samples and U6 RNA was used as an endogenous control. mir-195-5p sense primer was 5′- CGTAGCAGCACAGAAATATTGGC -3′.

Mir-195-5p mimics were designed and synthesized by the Gene Pharma Corporation (Shanghai, China), mir-195-5p was transfected into Huh7 and MHCC97L cells using lipo2000, and media were freshly replaced after 24 h.

### 4.10. Cell Proliferation Assay

A Cell Counting Kit-8 (CCK-8; Mei5bio, Beijing, China) was used to measure cell viability and evaluate cell proliferation. Ten microliters of CCK-8 solution were added to each well of an assay plate and incubated in a CO_2_ incubator for 3 h. The optical density was then detected at 450 nm using a microplate reader (Biotek, Vermont, USA).

### 4.11. Clone Formation Assay

The control cells and minis groups were digested and suspended, and 4000 cells were added to each 60 mm Petri dish. After culturing in a 37 °C incubator for 14 d, the cells were fixed with 4% paraformaldehyde, stained with crystal violet, and micrographed for counting.

### 4.12. Transwell Assay

Transwell chambers (Biofil, Guangzhou, China) were used for the cell invasion assays. The Matrigel-coated chamber was first placed in a 24-well tissue culture plate and left to stand for 2 h at 37 °C. The transfected control and minis group cells were trypsinized and suspended. Then, 500 μL of complete medium was added to the lower chamber, and 200 μL (8000 cells) of serum-free medium with cell suspension was added to the upper chamber. After 48 h of incubation, non-invading cells were wiped from the upper surface of the chamber membrane using a cotton swab. After fixing with 4% paraformaldehyde for 10 min, the cells on the lower surface of the chamber membrane were stained with crystal violet, counted, and images were taken under a microscope.

### 4.13. Statistical Analysis

Comparisons within the cell experiment were performed using Student’s t-test using GraphPad Prism Software (GraphPad Inc., California, USA). Statistical significance was set at *p* < 0.05.

### 4.14. FASN Functional Enrichment Analysis

Molecules with various interactions (physical interactions, co-expression, predicted, co-localization, genetic interactions, pathways, and shared protein domains) with FASN were analyzed using GeneMANIA, and functional enrichment analysis was performed. The GSEA method was used to evaluate the KEGG enrichment pathway of FASN in BLCA, LIHC, and PRAD in the LinkedOmics online database (http://www.linkedomics.org/admin.php).

## 5. Conclusions

According to our findings, FASN has a significant prognostic value for anti-PD-1 and anti-PD-L1 immunotherapy, which is negatively correlated with immune infiltration in the majority of tumors and has prognostic significance in a range of malignancies. Moreover, the expression of FASN may be regulated by DNA methylation, and DNA methylation of its promoter has clinical prognostic value. Finally, we proved that mir-195-5p may inhibit tumor activity via FASN inhibition, which has therapeutic drug development value.

## Figures and Tables

**Figure 1 ijms-23-15603-f001:**
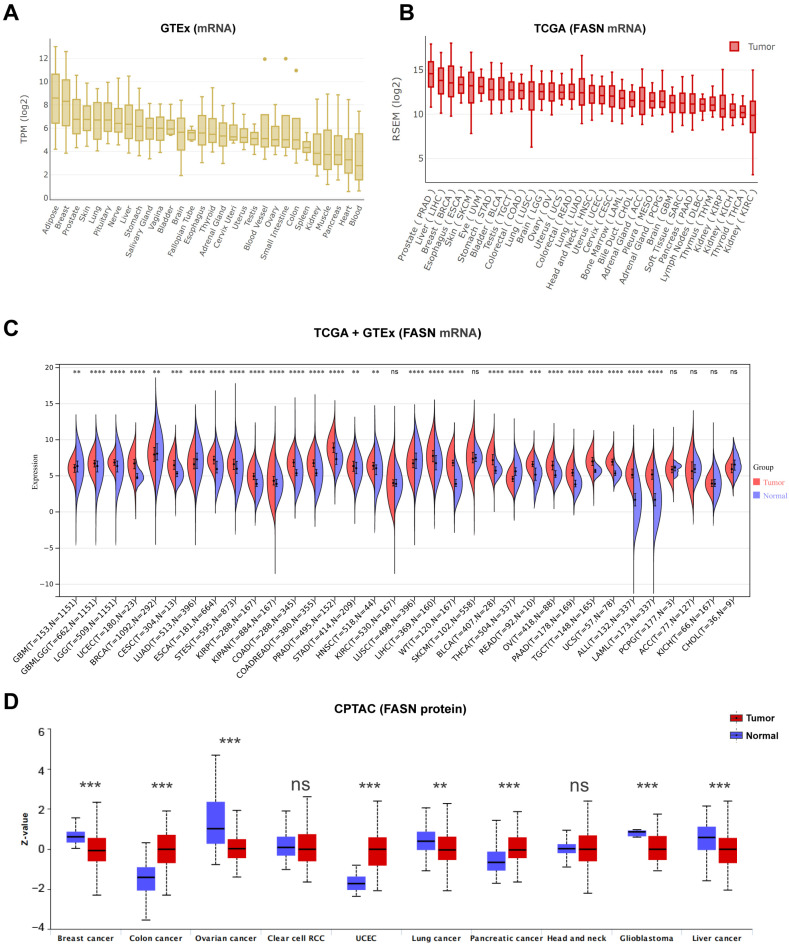
FASN expression profiles in normal tissues and cancers. (**A**) FASN expression levels in normal tissues and organs. (**B**) Profiles of FASN expression in 33 cancer types. (**C**) Sangerbox3.0 database indicated that the FASN expression in 34 cancer types. (**D**) The total protein level of FASN in normal tissue and breast cancer, colon cancer, ovarian cancer, clear cell RCC, UCEC, lung cancer, pancreatic cancer, head and neck cancer, glioblastoma, and liver cancer (** *p* < 0.01, *** *p* < 0.001, **** *p* < 0.0001, ns: *p* > 0.05.).

**Figure 2 ijms-23-15603-f002:**
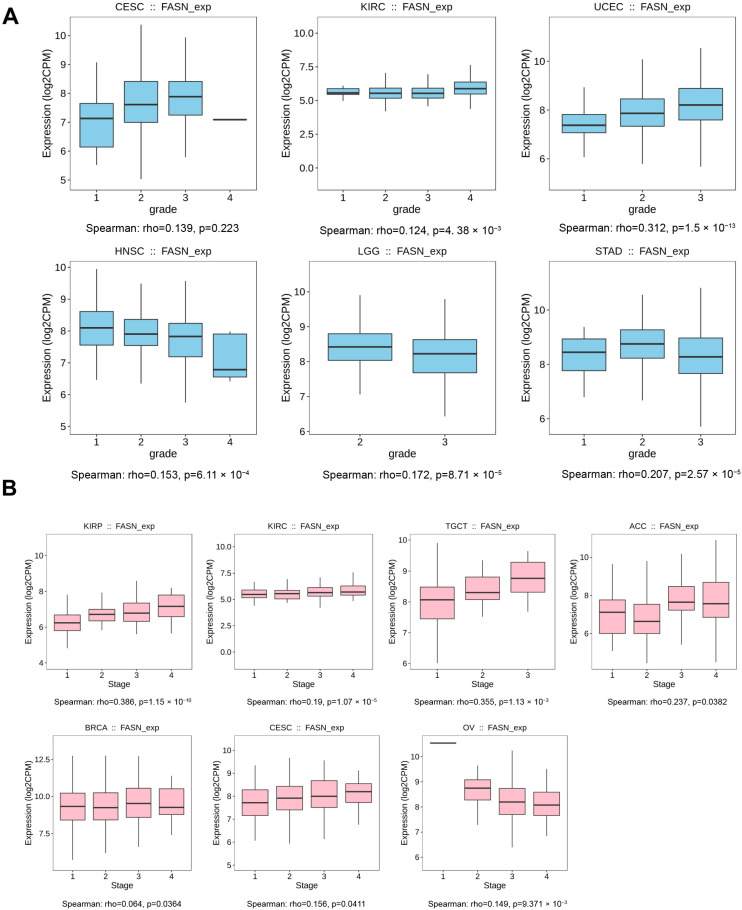
The correlation between FASN expression and cancer pathological grade. Grade (**A**) and stage (**B**) were analyzed by TISDB. FASN expression and progression grade and stages map with positive results are given (*p* < 0.05).

**Figure 3 ijms-23-15603-f003:**
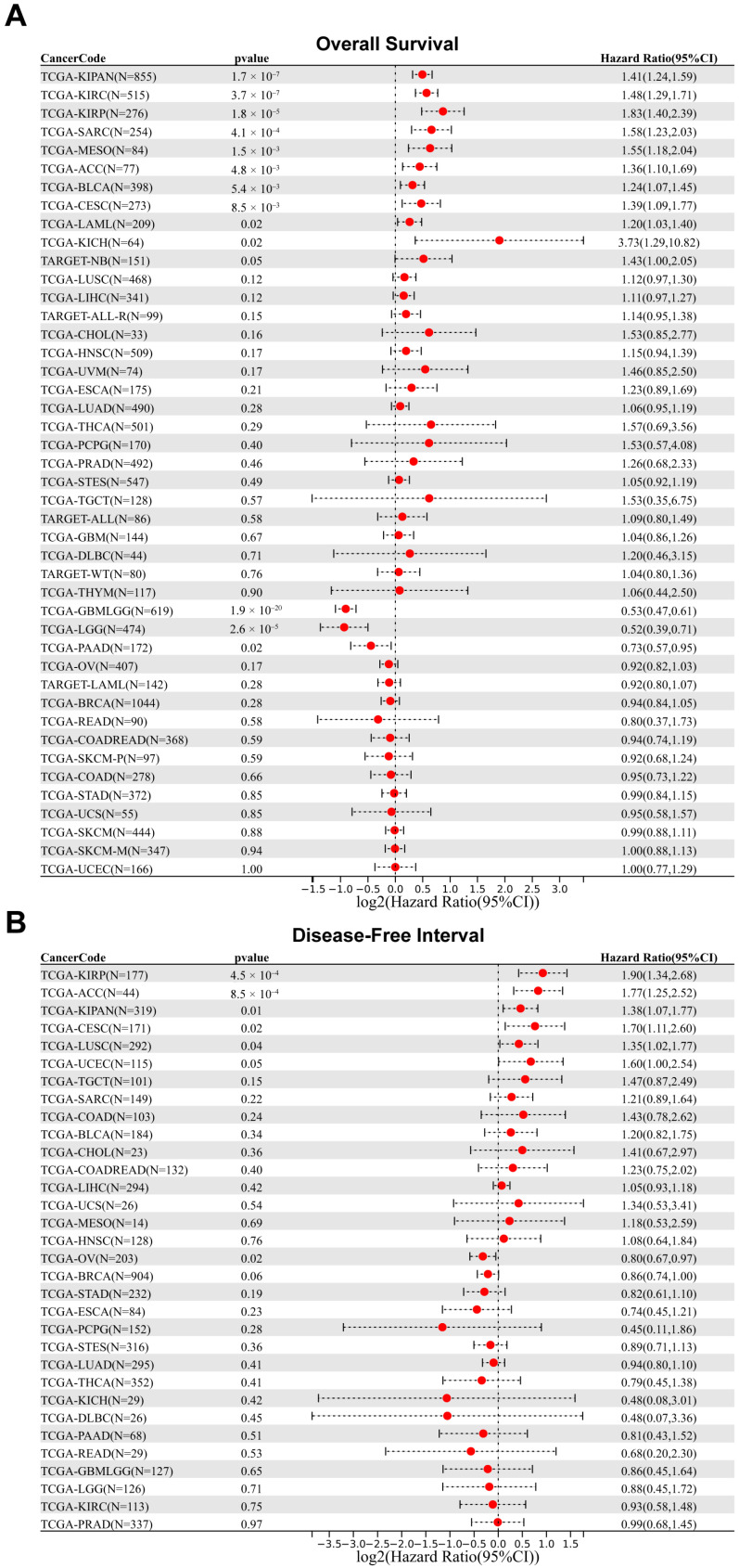
Correlation between FASN gene expression and survival prognosis of cancers in TCGA. We used the Sangerbox3.0 to perform overall survival (**A**) and disease-free interval (**B**) analyses of different tumors in TCGA.

**Figure 4 ijms-23-15603-f004:**
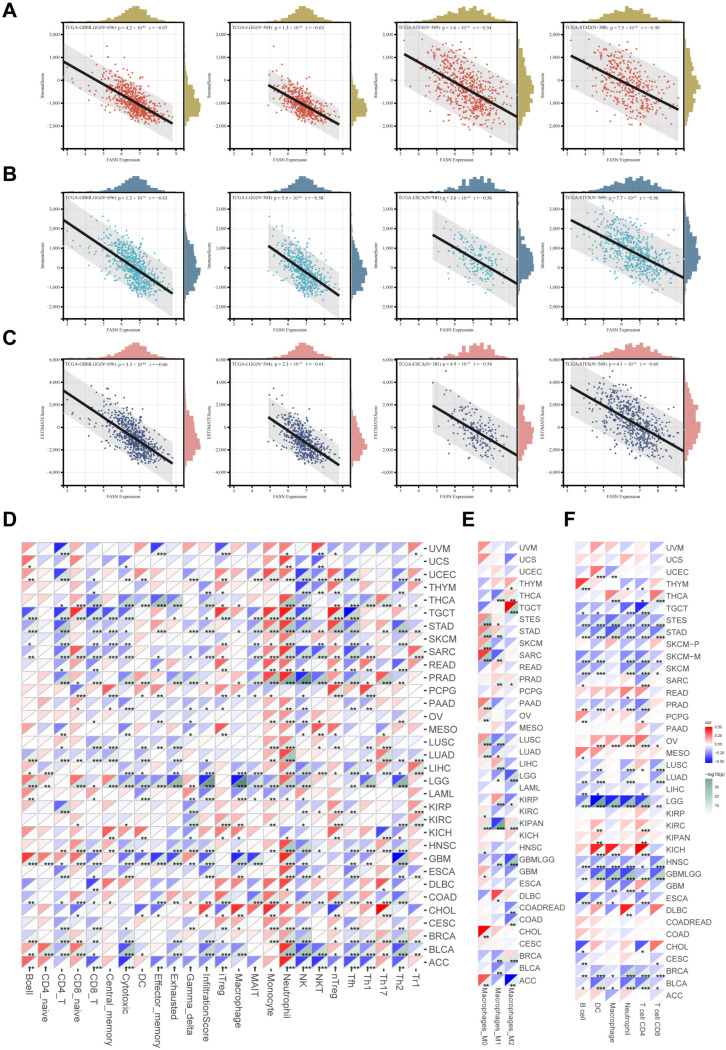
Correlations between FASN expression and immunity infiltration. (**A**) The top four tumors with the most significant correlation between the degree of immune infiltration and FASN expression were GBMLGG, LGG, STES, and STAD (StromalScore); (**B**) GBMLGG, LGG, ESCA, and STES (ImmuneScore); (**C**) GBMLGG, LGG, ESCA, and STES (ESTIMATEScore). (**D**) The correlation between FASN expression and 24 kinds of immune cell infiltration was analyzed by ImmuCellAI. (**E**) Sangerbox3.0 online website was used to analyze the relationship between FASN mRNA expression and M0 Macrophage, M1 Macrophage and M2 Macrophage infiltration. (**F**) Timer examined the correlation between FASN expression and six types of immune cell infiltration. (* *p* < 0.05, ** *p* < 0.01, *** *p* < 0.001.).

**Figure 5 ijms-23-15603-f005:**
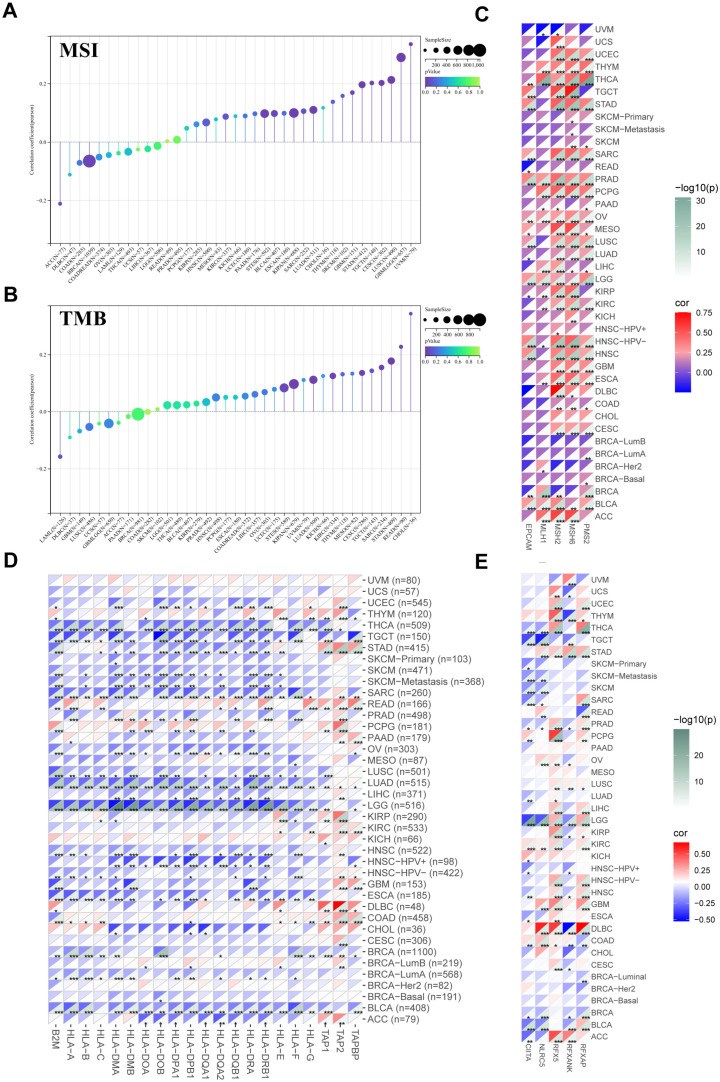
Correlation analysis of FASN expression with TMB, MSI, MMR-related genes, MHCs, and transcription factors. Sangerbox3.0 online website was used to analyze the relationship between FASN mRNA expression and MSI (**A**) and TMB (**B**). Using timer2.0 to analyze the correlation between FASN mRNA expression and MMRs (EPCAM, MLH1, MSH2, MSH6, PMS2) (**C**), MHCs (**D**), CITTA, RFX5, RFXANK, and RFXAP (**E**) in multiple cancers. (* *p* < 0.05, ** *p* < 0.01, *** *p* < 0.001.).

**Figure 6 ijms-23-15603-f006:**
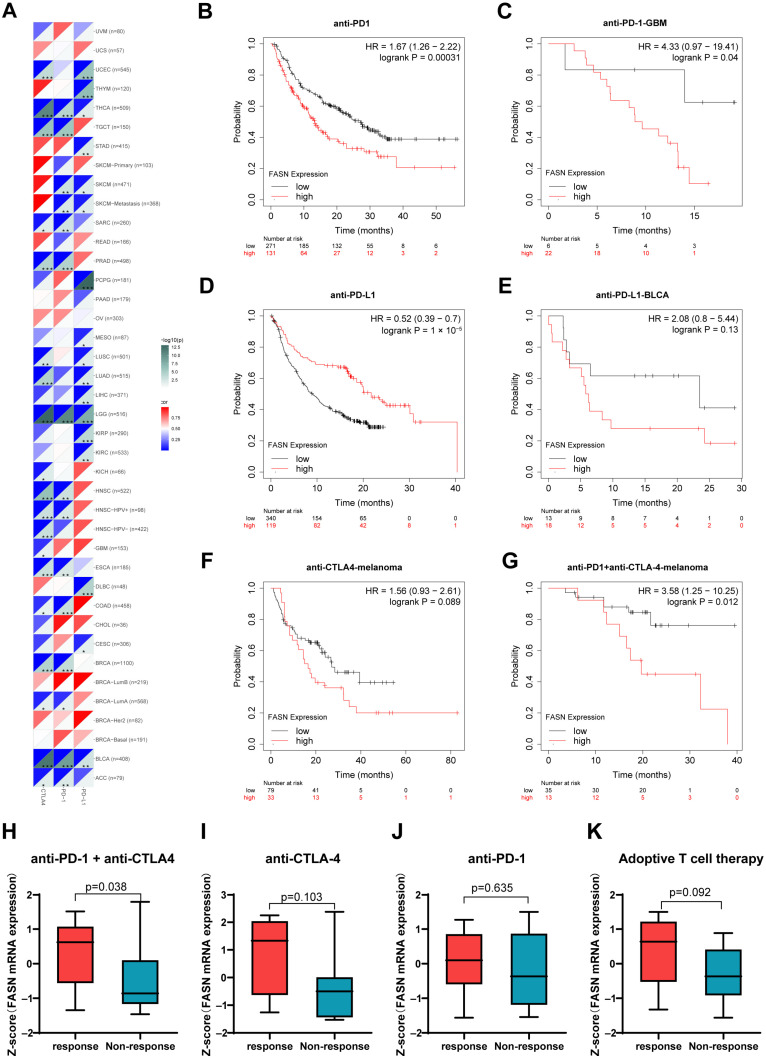
Correlation analysis of FASN expression with PD-1, PD-L1, and CTLA-4 and differential analysis of the two indicators to the responsiveness of immunotherapy. (**A**) Correlation of FASN expression with PD-1, PD-L1, CTLA-4 in pan-cancer. (* *p* < 0.05, ** *p* < 0.01, *** *p* < 0.001.). Prognostic value of FASN expression in patients with different cancers treated with anti-PD-1, (**B**,**C**) anti-PD-L1 (**D**,**E**), anti-CTLA-4 (**F**), anti-PD-1 combined with anti-CTLA-4 (**G**), respectively. Correlation of FASN expression with four immunotherapy responses of anti-PD-1 combined with anti-CTLA-4 (**H**), anti-CTLA-4 (**I**), anti-PD1 (**J**), and adoptive T cell therapy (**K**).

**Figure 7 ijms-23-15603-f007:**
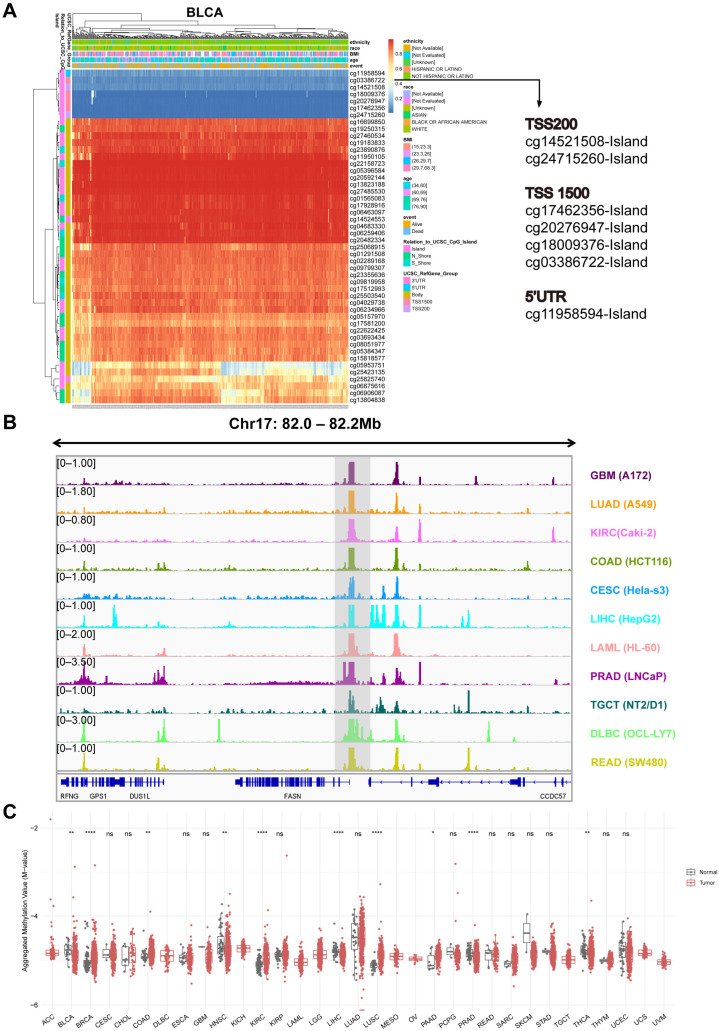
DNA methylation analysis of FASN promoter. (**A**) Distribution of DNA methylation at the FASN locus in the representative cancer BLCA. (**B**) DNase-seq data in eleven cancers obtained from the ENCODE database were visualized using the integrative genomics viewer, coordinates were located at the FASN locus, and the peaks of the target are shown in shaded areas. (**C**) Differences in aggregated methylation of seven methylation probes (cg24715260, cg14521508, cg17462356, cg20276947, cg18009376, cg03386722, cg11958594) of FASN in multiple cancers. (* *p* < 0.05, ** *p* < 0.01, **** *p* < 0.001).

**Figure 8 ijms-23-15603-f008:**
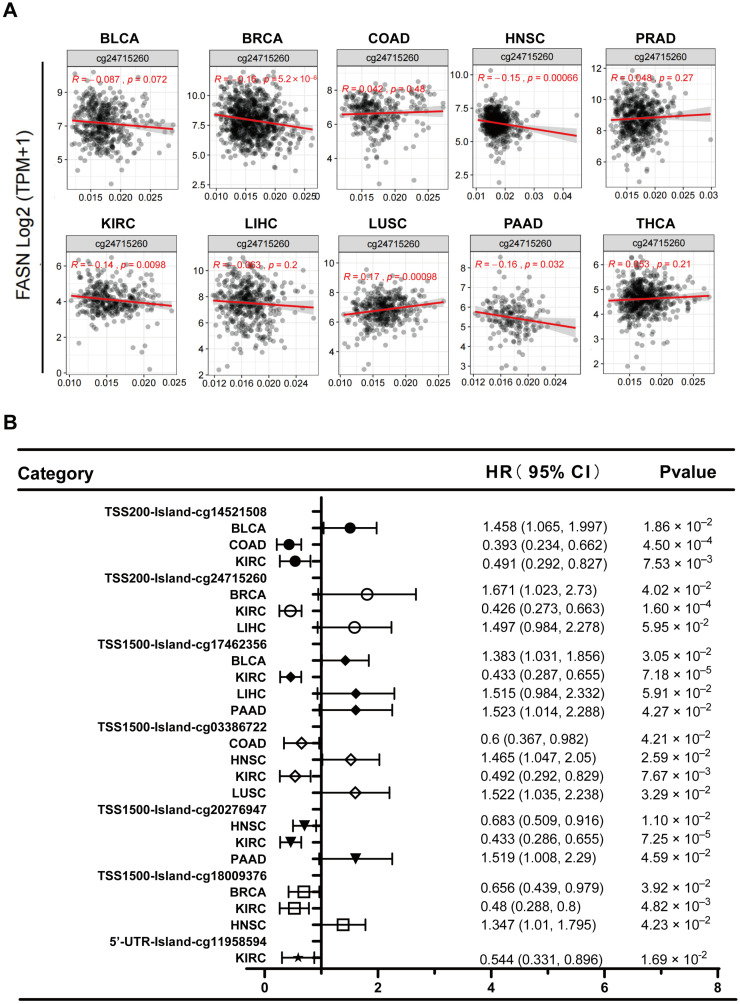
Correlation analysis of FASN promoter methylation and expression and prognostic analysis of seven FASN methylation probes in ten cancer types. (**A**) Correlation of DNA methylation at the cg24715260 probe position with FASN expression in ten tumors. (**B**) Prognostic analysis of seven probes in ten cancers with positive results (*p* < 0.05) forest plots are given.

**Figure 9 ijms-23-15603-f009:**
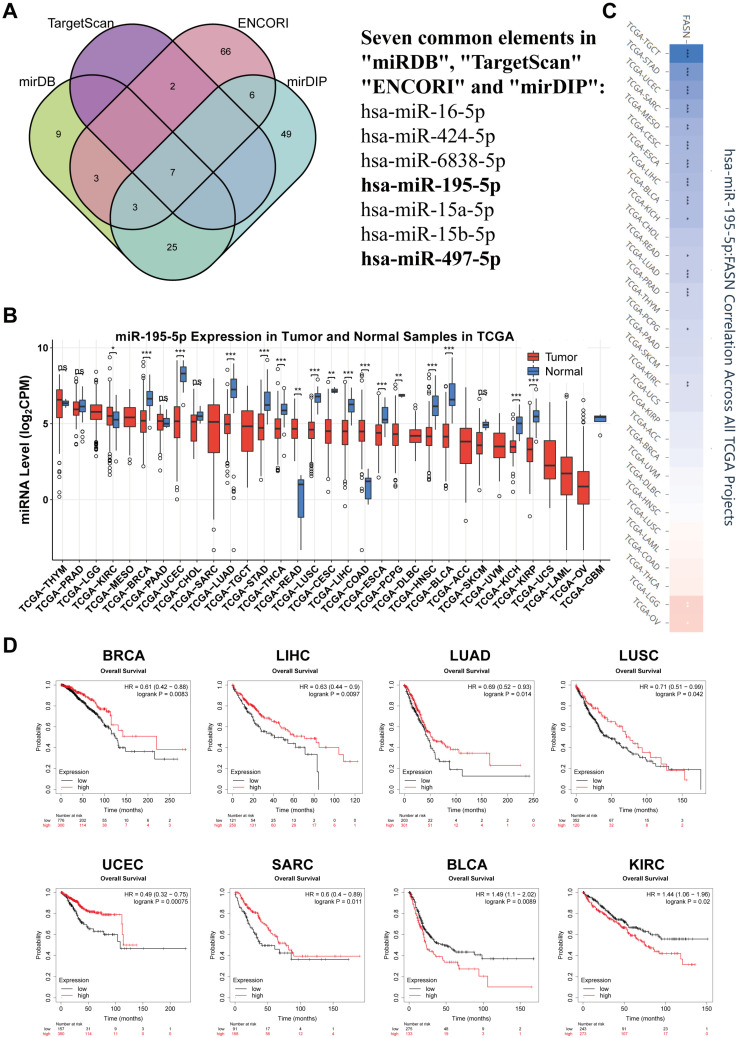
Mir-195-5p is a potential regulatory gene of FASN. (**A**) mirDB, TargetScan, ENCORI, and mirDIP four databases were used to analyze miRNAs that may target FASN, and obtain seven conserved miRNAs by overlap. Expression profile of mir-195-5p in different cancers (**B**) and correlation with FASN (**C**) analyzed with CancerMIRNome. (* *p* < 0.05, ** *p* < 0.01, *** *p* < 0.001, ns: *p* > 0.05.). (**D**) Kaplan–Meier plotter analysis of the correlation between the expression of mir-195-5p and prognosis in multiple cancers. The survival map and Kaplan–Meier curves with positive results are given.

**Figure 10 ijms-23-15603-f010:**
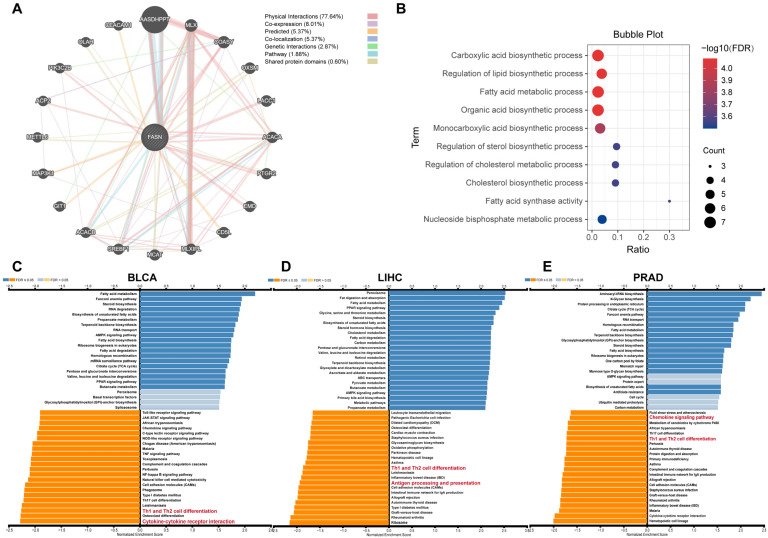
FASN interacting molecules and functional enrichment analysis. Analysis of various molecules (**A**) interacting with FASN and functional enrichment analysis (**B**) of FASN using GeneMANIA. FASN enrichment KEGG pathways in BLCA (**C**), LIHC (**D**), and PRAD (**E**) were analyzed by GSEA.

## Data Availability

The datasets analyzed for this study can be found in the online repository included in the materials and methods section. Further inquiries can be directed to the corresponding authors.

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
