# Peer review of "Comprehensive Analysis of FASN in Tumor Immune Infiltration and Prognostic Value for Immunotherapy and Promoter DNA Methylation"

_ijms, 2022, doi:10.3390/ijms232415603_

Round 1

Reviewer 1 Report

In this manuscript, Zhang et al. investigated the roles of FASN in tumor-infiltrating immune cells and highlighted the promising prognostic value of FASN for immunotherapy. Although the lack of experiment validation might weaken the reliability of conclusion, large amounts of bioinformatics data provide research directions and basis for future preclinical and clinical studies. I only have some minor suggestions:

1. Figure 4D, the authors should clarify different immune cell populations. There are some misleading labels for the cell types. For example, CD8+ T cells are cytotoxic T cells, and CD4+ T cells include Th1, Th2 and Th17.

2. Almost all MHCs were negatively correlated with FASN expression. The authors claimed that FASN might be related to CIITA. However, CIITA is the transcription factors for the MHC-II molecules, not for MHC-I. Figure 5D shows the negative correlation of B2M, HLA-A,B,C,E,F,G, and these are all MHC-I molecules. The authors should discuss the relationship between FASN and MHC-I.

3. There are some typos in this manuscript. The authors need to proofread and correct.

Line 145, up-regulation.

For the P value, there should be space before and after < and =. For example, it should be P < 0.05, instead of P<0.05. It should be p = 8.5e-3, instead of p=8.5e-3.

Line 236, CIITA

Reviewer 2 Report

Dear Authors, 

thank you for this interesting paper. I hope the following comments will help you to improve the manuscript.

Line 12: pan-cancer - what does it mean? Might be incomprehensible for some younger readers

Line 16: significantly related - not sure it this expression is appropriate, English language correction recommended in entire paper; there are many places that should be significantly improved, especially when it comes to grammar

Line 22: miR-195-5p targets FASN or its effect affects FASN?

line 82: GTEx abbr. used for the first time, please explain, some users may not be familiar with it. 

Figure 3: very small tables/letters, difficult to read. Can you make them bigger? Maybe rearrange the graphs?

Figure 4: as above, cannot read

Figure 10: as above, cannot read

There are also some papers you may find useful, and I can't see them in your reference section, for example:

https://pubmed.ncbi.nlm.nih.gov/33806327/

https://www.ncbi.nlm.nih.gov/pmc/articles/PMC6200091/

https://www.nature.com/articles/s41437-022-00533-1 

https://pubmed.ncbi.nlm.nih.gov/31869744/
